# Disinfectants against African Swine Fever: An Updated Review

**DOI:** 10.3390/v14071384

**Published:** 2022-06-24

**Authors:** Maria Serena Beato, Federica D’Errico, Carmen Iscaro, Stefano Petrini, Monica Giammarioli, Francesco Feliziani

**Affiliations:** National Reference Laboratory for Asfivirus and Pestivirus, Istituto Zooprofilattico Sperimentale dell’Umbria e delle Marche, “Togo Rosati”, Via G. Salvemini 1, 06126 Perugia, Italy; f.derrico@izsum.it (F.D.); c.iscaro@izsum.it (C.I.); s.petrini@izsum.it (S.P.); m.giammarioli@izsum.it (M.G.); f.feliziani@izsum.it (F.F.)

**Keywords:** African Swine Fever, disinfectant, review

## Abstract

African Swine Fever (ASF), a hemorrhagic disease with a high mortality rate in suids, is transmitted via direct and indirect contact with infectious animals and contaminated fomites, respectively. ASF reached Europe in 2014, affecting 14 of the 27 EU countries including, recently, the Italian peninsula. The fast and unprecedented spread of ASF in the EU has highlighted gaps in knowledge regarding transmission mechanisms. Fomites, such as contaminated clothing and footwear, farming tools, equipment and vehicles have been widely reported in the spread of ASF. The absence of available vaccines renders biosecurity measures, cleaning and disinfection procedures an essential control tool, to a greater degree than the others, for the prevention of primary and secondary introductions of ASF in pig farms. In this review, available data on the virucidal activity of chemical compounds as disinfectants against the ASF virus (ASFV) are summarized together with laboratory methods adopted to assess the virucidal activity.

## 1. Introduction

African Swine Fever (ASF) is a hemorrhagic disease affecting all ages of domestic and wild pigs, showing very high mortality rates during infection with high virulent strains, and it is caused by a large cytoplasmatic linear double-stranded DNA virus. It is currently classified as the sole member of the family *Asfarviridae*, genus *Asfivirus* [1,2]. The ASF virions are icosahedral and approximately 200 nm formed by concentric layers: the internal core, the core shell, the inner membrane, the capsid and, in the extracellular virions, the external envelope [3,4,5]. The outer envelope derives from the cellular plasma membrane during the budding process by which ASF virus (ASFV) egresses from the cell [4,5]. Based on the B646L gene in the C-terminal end of the ASFV genome, 24 genotypes are identified [6,7,8,9,10,11] and other viral genes have been used for genotyping [12,13,14,15]. All the genotypes have been detected in the African continent [11,16,17,18].

ASFV has become possibly the most relevant epizootic disease from an animal health perspective. After the introduction of genotype II strain in Georgia in 2007, the virus reached the Russian Federation and moved westwards, invading the European Union (EU) in 2014 [19,20]. Four years later, ASFV was detected in China and soon after, in several Asian countries [21,22]. The rapid spread of ASFV over the last decade has raised great concern related to consequences for animal health and for the direct and indirect economic impacts. This concern is increasing dramatically with the actual confirmation of ASFV in fourteen out of 27 EU countries [23] and, recently, in wild boars and domestic pigs in the Italian peninsula. The high risk of the introduction of ASFV to non-affected EU countries has intensified scientific efforts to address a great number of questions that have been raised with the goal of finding proper actions and strategies to combat the ongoing ASFV issue. Susceptible suids can be infected by direct or indirect contact with infectious animals or their fluids (oral and nasal exposure and cutaneous wounds), ingestion of contaminated animal feed, pork or pig products, or contact with contaminated surfaces and materials present in the environment and fomites acting as mechanical vectors of the disease. In addition, soft ticks of the genus *Ornithodoros* [24] act as competent vectors [25,26]. Fomites, such as contaminated clothing and footwear, farming tools, equipment and vehicles have been widely reported as responsible for ASFV spread. Fomites and human-related activities therefore have a crucial role in ASFV transmission in domestic and wild pigs. In addition, the carcasses of animals (i.e., wild boars) that succumbed to diseases may be considered as pathogen reservoirs, posing additional challenges for the control of the infection. Furthermore, the absence of available vaccines render biosecurity measures an essential control tool, more than the others, for the prevention of primary and secondary introductions of ASFV in pig farms. In this context, the availability of data regarding the virucidal activity of chemical compounds against ASFV provide useful information to properly plan decontamination procedures aimed at preventing the introduction of ASFV and to limit its secondary spread. The present review’s objective was to summarize information on chemical substances tested against ASFV for their virucidal activity and to provide information on specific situations in which the use of a given product is recommended or discouraged. The information reported should supply the reader with concise and practical information on how to develop specific sanitation programs aimed at the ASFV inactivation. The cleaning and disinfection procedures in pig holdings against ASFV have been reviewed elsewhere [27].

## 2. Results

Based on their resistance to chemical agents, viruses were classified into three different categories by Holl and Youngner [28], namely A, B and C. Such classification is based on two virus characteristics: (1) presence/absence of lipids and (2) size, which determines the virus’ susceptibility to disinfectants. Category A viruses are an intermediate to large size, contain lipids and are very susceptible to detergents, soaps, all the disinfectants and dehydration and often do not persist long unless the environment is moist and cool. Category B viruses are smaller in size, with no lipid membrane, and they are relatively resistant to lipophilic disinfectants, such as detergents. Category C viruses (e.g., adenoviruses and reoviruses) are intermediate in size with no lipids; they are intermediate between categories A and B in sensitivity to the best virucidal disinfectants, such as hypochlorite, alkalis, oxidizing agents and aldehydes. The ASFV belongs to category A.

### 2.1. Methods for Testing the Virucidal Activity of Disinfectants

Several protocols have been established to test the efficacy of disinfectants against viruses under different conditions. In the present section, methods adopted to investigate the efficacy of chemical compounds specifically against ASFV are described and graphically represented. Some of these protocols have been issued as standards by international bodies in the EU (i.e., UNI EN 14675:2015) and the United States of America (ASTM E1053-20) [29,30] and by the Organization for Economic Co-operation and Development (OECD) as guidelines [31]; others have been developed and modified from these ones to specifically investigate the efficacy of chemical products against ASFV [32,33,34]. In the EU, methods for testing the efficacy of disinfectants and antiseptics have been being developed by the Technical Committee 216 (TC216) of the European Committee for Standardization (CEN) since 1989 [35,36,37]. These methods foresee on a three phases of testing disinfectants and antiseptics. Briefly, phase 1 (suspension methods) determines the bactericidal, fungicidal, yeasticidal or sporicidal activity without regard to the specific areas of application; phase 2/step 1 tests (quantitative suspension methods); phase 2/step 2 tests are based on the carrier method; phase 3 methods are intended to test the product under the practical in-use conditions and currently, there are no drafts or standards available. Some standards are specific to the veterinary area, i.e., UNI EN 14675:2015 corresponds to the quantitative suspension method (phase 2/step 1) [29] and the UNI EN 17122:2020 corresponds to the carrier method (phase 2/step 2) [38]. The test methods (standards or guidelines) available to test the virucidal activity of chemical compounds and adopted for the ASFV are distinguished into: suspension (i.e., UNI EN 14675:2015, phase 2/step 1) [29] and carrier tests (i.e., OECD guidelines and ASTM E1053-20). The suspension test is based on the contact of the cell-cultured ASFV with the disinfectant in liquid form. By contrast, in the carrier test, the virus is spotted on a surface, dried and subsequently exposed to a disinfectant, either sprayed or put in contact in its liquid form. In addition, other variables such as the virus−disinfectant ratio, the compulsory temperature and contact time and the presence and type of interfering substances, constitute crucial differences/variables among test methods (Table 1). The methods used to test the virucidal activity of chemical compounds against the ASFV share similar steps with some differences that are presented in Table 1.

The common goal of test methods applied to verify the virucidal activity of a chemical compound is to prove a virus titer reduction of three or four log_10_, according to the test method, following the physical contact between the virus and the chemical compound/disinfectant undergoing testing. Therefore, a high ASFV titer is required (e.g., 5.5–6.5 log_10_ TCID_50_/mL) to demonstrate the required virus titer log_10_ reduction and, ultimately, the efficacy of the chemical compound.

Briefly, chemical compounds to be tested are diluted with water of a standardized hardness (Table 1). The suspension test according to the UNI EN 14675:2015 standard indicate interfering substances: bovine serum albumin (BSA) (3.0 g/L) simulates a low-level soiling condition and a higher concentration of BSA (10 g/L) plus yeast extract (YE) 10 g/L simulates a high-level soiling condition. One part of the virus suspension is mixed with one part of the interfering substance and incubated at +10 °C for 2 min. Subsequently, eight parts of the chemical diluted to 1.25-fold of each tested concentration is added. The obtained mixture of the virus, tested chemical and interfering substance is incubated at +10 °C for 30 min, identified as compulsory test conditions. Afterwards, test tubes are placed on crushed ice to stop the virucidal activity of the tested product. Samples are then immediately serially diluted 10-fold (in replicates) to perform virus titration on a cell culture and assess the residual virus titer post contact with the disinfectant (Figure 1). Additional temperatures and contact times to the compulsory ones (+10 °C and 30 min) can be tested according to the application of the chemical compound.

Regarding the carrier test methods, the ASTM E1053-20 standard and the OECD guidelines are specific for non-porous surfaces. Both methods were slightly modified for ASFV studies, as described below [32,34,39].

The ASTM E1053-20 standard is adopted for chemicals that must be registered to the Environmental Protection Agency (EPA) in the US and provides indications on how to test a disinfectant on non-porous environmental surfaces, such as glass surfaces; specifically, glass petri dishes must be used (Figure 2). Krug and collaborators adopted this method with some modifications to assess the virucidal activity of disinfectants against the ASFV on non-porous surfaces [32,34]. In particular, for non-porous surfaces, they used a plastic surface represented by six-well cell culture plates, stainless steel mold and sealed concrete instead of glass petri dishes (Figure 2), and they used a different virus−disinfectant ratio: 1:5 for stainless steel and plastic surfaces.

Briefly, the virus is mixed with an interfering substance, fetal bovine serum (FBS) at 5%, in a ratio of 1:10 (*v*/*v*). This mixture is spotted on the non-porous surface and dried at room temperature (+20–25 °C). Subsequently, the disinfectant is added for a contact time indicated on the disinfectant label. In order to stop the disinfectant activity, a specific neutralizer is added and virus titration is carried out to prove the four log_10_ virus titer reduction.

The OECD guidelines were issued in 2013 to evaluate the virucidal activity of compounds specifically on hard non-porous surfaces (Figure 3).

The OECD guidelines indicate to use stainless steel disks as hard non-porous surfaces. The steps are similar to the ASTM standard: the preparation of the virus−disinfectant mixture with the addition of BSA, YE and bovine mucin (BM) as interfering substances; this is an inoculation step on the selected non-porous surface and a step to dry the mixture on the surface. Following the drying step, the disinfectant is applied in a liquid form at the temperature and contact time indicated on the disinfectant label (Figure 3, Table 1). The recovery of the viable virus is performed after the addition of a specific neutralizer by vortexing the disinfected−infected surface. Only one study of the ASFV by Gabbert et al. [39] adopted this method with slight modifications using carbonated concrete coupons as porous surfaces instead of the stainless steel disks indicated as the reference non-porous surface according to the OECD guidelines [31]. Other studies investigated the virucidal activity of disinfectants against the ASFV without adopting any international standards [40,41,42,43,44]. These studies tested the virucidal activity resembling the UNI EN 14675:2015 standard by putting the disinfectant in the liquid form and the cell-cultured ASFV strain in contact.

### 2.2. Chemical Compounds Tested against ASFV

Disinfectants active against the ASFV can be grouped into eight categories: acids, alkalis, aldehydes, chlorine and chlorine compounds, iodine compounds, oxidizing agents, phenol compounds and quaternary ammonium compounds (QACs). In addition, recently, several plant extracts were tested against the ASFV. A list of tested and efficacious disinfectants against the ASFV, according to studies available in the literature, is reported in Table 2.

#### 2.2.1. Acids

Acid compounds are distinguished as organic or inorganic. Both categories are able to inactivate viruses through the decrease in pH values and organic ones by the interaction of lipophilic structures with the membranes of enveloped viruses [53]. Inorganic acids, with exception of citric acid, have limitations in their use due to their corrosiveness [54]. Citric acid (C_6_H_8_O_7_) is an organic compound available in solid form that can be safely used for personal use and clothing decontamination [55].

Krug et al. 2011 and 2012 [32,33] tested the efficacy of citric acid (C_6_H_8_O_7_) against the BA71V strain on porous (birch wood) [33] and non-porous surfaces (steel and plastic surfaces) [32]. When testing the efficacy of citric acid (2%) on a porous surface (birch wood veneer), Krug et al. 2012 [33] demonstrated the ability to reduce by four log_10_ an initial ASFV titer of 10^8.3^ TCID50/mL following 20 min of contact time. In this experiment, they observed a fast ASFV titer decrease within the first 10 min and after 30 min of exposure to 2% citric acid (C_6_H_8_O_7_), the ASFV titer was below the detection limit [33]. Citric acid (C_6_H_8_O_7_) was tested on a non-porous surface (plastic) as well at 1 and 2% concentrations for 10 min and both concentrations resulted in a four log_10_ reduction of the ASFV titer [32]. One percent citric acid (C_6_H_8_O_7_) resulted in a four log_10_ reduction of ASFV on the steel surface as well [32]. The method by Krug et al. 2011 [32] adopted to test the virucidal activity of the citric acid (C_6_H_8_O_7_) on porous and non-porous surfaces was also adopted to the ASFV spiked into the swine blood and feces dried on stainless steel coupons [34]. The presence of blood greatly affected the efficacy of citric acid (5%) compared to its ability to induce a titer reduction of a cell-cultured ASFV under the detection limit in 8 min [34]. By contrast, 2% citric acid rapidly (in 2 min) inactivated the dried ASFV-infected swine feces on the stainless steel surface [34]. More recently, Juszkiewicz et al. [45] tested three different concentrations of acetic acid (C_2_H_4_O_2_) (1, 2 and 3%) against the Vero-adapted BA71V ASFV strain. According to the method used (UNI EN 14675:2015), only the higher concentrations (3% and 2%) proved to decrease the virus titers of four log_10_ in low-level soiling conditions (i.e., in the presence of BSA) and only 3% acetic acid induced a four log_10_ ASFV titer reduction in high-level soiling conditions (i.e., BSA + YE).

Two percent acetic acid (C_2_H_4_O_2_) and citric acid (C_6_H_8_O_7_) proved to be efficacious against the same ASFV strain (BA71V) in two different studies applying two different test methods (Table 2), suggesting that at this concentration, they might be used in field conditions.

In conclusion, the data available show that acids might be adopted for disinfection purposes against the ASFV being safe for personnel and particularly indicated (citric acid) for the decontamination of clothes.

#### 2.2.2. Alkalis

Sodium hydroxide (caustic soda, NaOH) and sodium carbonate (washing soda, Na_2_CO_3_) are the major representative compounds of this group of chemicals. They maintain their disinfectant properties in the presence of heavy burdens of organic material. They are easily available at a low cost and are extensively used as disinfectants in the food and dairy industry; they are indicated to be efficacious agents for the decontamination of animal housing and associated structures [55]. However, it should be considered that contact with very high concentrations of sodium hydroxide (NaOH) could cause severe effects to the eyes, skin, digestive system or lungs, resulting in permanent damage or death. Prolonged or repeated skin contact may cause dermatitis. Repeated inhalation of sodium hydroxide vapor can lead to permanent lung damage.

Turner et al. [40] tested granular sodium hydroxide (NaOH), and powdered calcium hydroxide (Ca(OH)_2_). The ASFV strain used was the Lilongwe 20/1 from Malawi [40]. NaOH, Ca(OH)_2_ and the ASFV were added to pig slurry to obtain 10% *v*/*v* of the virus−disinfectant mixture. The efficacy of NaOH and Ca(OH)_2_ was assessed at two different temperatures: +4 °C and +22 °C. The chemicals were tested in different concentrations: NaOH: at 1, 0.5, 0.2 and 0.1% (*w*/*v*) and Ca(OH)_2_ at 2, 1, 0.5 and 0.2% (*w*/*v*) for 30 min. Ca(OH)_2_ was effective in inactivating the ASFV at 1% and NaOH at a lower concentration of 0.5% when tested in MEM at both temperatures. Both chemicals were effective against the ASFV at 1% at +4 °C but not at +22 °C. When both chemicals were tested at 0.5% at +4 °C, no virucidal effect was observed against the ASFV. Within 30 min, Ca(OH)_2_ was able to inactivate the ASFV at 1 and 0.5% at +4 °C, and +22 °C NaOH was effective from 1 to 0.2% at 22 °C and at 1 and 0.5% at 4 °C.

Kalmar et al. [44] tested the effect of alkalinization on the ASFV strain Lisbon 60, pre-diluted in MEM or porcine plasma. The effect of alkalinization was studied by adding NaOH at different concentrations to obtain a pH value of 10.2. NaOH was added at 0.042% in MEM and at 1% in porcine plasma. The effect of alkalinization was tested in combination with heat treatment and the ASFV was consistently inactivated at +48 °C and a pH of 10.2 within 25 min from a starting titer of 10^3.51^ TCID_50_/mL. In the same conditions, the sensitivity of the ASFV diluted in porcine plasma was lower and the ASFV remained viable following a 60 min treatment, although a reduction in the virus titer was observed [44].

Caustic soda (NaOH) was also tested recently at three different concentrations: 1, 2 and 3% against the ASFV BA71V strain according to the European Standard UNI EN 14675:2015 using low-level soiling conditions (BSA) and high-level soiling conditions (BSA + YE) for 30 min of contact time [45]. All concentrations were effective in the low- and high-level conditions, except for the lowest dilution at high-level conditions.

Krug and collaborators tested the efficacy of sodium carbonate (Na_2_CO_3_) against the BA71V strain on non-porous surfaces (steel and plastic) [32]. Briefly, the ASFV was dried on steel and plastic and then exposed to 4% sodium carbonate (Na_2_CO_3_) for 30 min at +22 °C. At the end of the contact time, the virus−disinfectant mixture was tested for virus viability by virus titration on cell cultures. The sodium carbonate (Na_2_CO_3_) (4%) did not inactivate (four log_10_ reduction) the ASFV [32].

Collectively, these studies indicated that NaOH (caustic soda) is effective against the ASFV at a 1% concentration following 30 min of contact time. Regarding calcium hydroxide (Ca(OH)_2_) and sodium carbonate (Na_2_CO_3_), only one study is available against the ASFV; therefore, indications on their use against the ASFV cannot be provided.

#### 2.2.3. Aldehydes

Aldehydes are organic compounds that cause virus inactivation through the alkylation of amino and sulphydrilic groups of proteins and purine bases [54]. The virucidal activity is based on the alkylation of amino and sulphydrilic groups of proteins and purinic bases. The efficacy of aldehydes decreases in the presence of organic matter. Glutaraldehyde is the most used compound for disinfection purposes, although precaution should be adopted to avoid irritation of the eyes and skin. The first study available on the virucidal efficacy of aldehydes against the ASFV was from Cunliffe et al. [56] testing glutaraldehyde (C_5_H_8_O_2_). This study’s objective was to investigate the safety of glutaraldehyde-fixed swine aortic valve bioprostheses. Aortic valves from experimentally ASFV-infected swine were exposed to glutaraldehyde in the following way: aortic valves were immersed in 0.2% buffered glutaraldehyde and stored at room temperature for 24 h. The buffered glutaraldehyde was replaced by a fresh one (0.2%) and the valves were exposed for another 10 days. Following this contact time, the valve suspension was used to inject intramuscularly swine (*n* = 4) in order to assess the virus viability. None of the swine injected with the valve suspension developed clinical signs and antibodies 14 days post-inoculation. Juszkiewicz et al. [45,48] reported the efficacy of glutaraldehyde on the ASFV. In the first study performed, a glutaraldehyde (C_5_H_8_O_2_) solution in water (25% CAS: 111-30-8) was tested against the ASFV BA71V using the suspension method (UNI EN 14675:2015) at three different concentrations: 0.1, 0.5 and 1% at both soiling conditions [45]. At low but not at high-level soiling conditions, glutaraldehyde solution was effective at all concentrations tested in yielding a four log_10_ virus titer reduction. The same group tested a commercially available product based on glutaraldehyde using the same test method [29]. This study evidenced a cytotoxic effect of a glutaraldehyde (C_5_H_8_O_2_)-based product for cell cultures and, therefore, it turned out to be untestable with the selected test method. Regarding the virucidal activity of formaldehyde (CH_2_O) against the ASFV, only one study is available [45] that applied the suspension test according to the EU standard and demonstrated its cytotoxicity effect for cell cultures at all dilutions tested. Therefore, no data are available on the efficacy of formaldehyde against the ASFV. However, the toxicity for humans from such a compound makes it the very last choice for any disinfection procedures.

In agreement with general knowledge on the use of glutaraldehyde (C_5_H_8_O_2_) as a disinfectant, the available studies on its activity against the ASFV indicate that it is efficacious at 1% for 30 min of contact time.

#### 2.2.4. Chlorine and Chlorine Compounds

This group of compounds based their disinfectant activity on the oxidation of peptide links, denaturizing proteins [57]. The disinfectant property of this group is due to the hypochloric acid (HOCl) produced at an acidic pH. Compounds derived from chlorine are widely used and available in both liquid (sodium hypochlorite, NaClO) and solid forms (calcium hypochlorite, Ca(ClO)_2_). They are inexpensive and rapidly efficacious but corrosive and inhibited by organic material, and their stability is pH dependent: at high pH values, this group of disinfectants show a decrease in their efficacy [54].

The virucidal activity of sodium hypochlorite (NaClO) was tested against the BA71V ASFV strain using the so called “suspension method” according to the UNI EN 14675:2015 at 0.3, 1 and 1.5% in low- and high-level soiling conditions. All concentrations resulted in a four log_10_ reduction of ASFV in the low-level soiling condition; by contrast, only 1 and 1.5% reduced the virus titer of four log_10_ in the high-level soiling condition [45].

Sodium hypochlorite (NaClO) was tested for its efficacy in inactivating the ASFV on a porous surface, such as birch wood veneer, at the concentration of 1000 parts per million (ppm) according to the ASTM E1053-20 standard with a few modifications (Figure 2). Following a contact time of 20 min, viable ASFV was still detectable [33]. Doubling the concentration of the sodium hypochlorite to 2000 ppm, a complete inactivation of the ASFV was achieved within 30 min [33]. Sodium hypochlorite at 500 ppm concentration was tested on steel and plastic surfaces to mimic the non-porous surface conditions. Following 10 min of contact time, 500 ppm sodium hypochlorite caused the ASFV titer reduction of four log_10_ [32].

Sodium hypochlorite (NaClO) (1500 ppm) was tested to determine its ability to inactivate ASFV spiked into swine blood and feces and dried on stainless steel coupons [34] according to the ASTM 1053-20 method (Figure 2). The tested concentration of sodium hypochlorite (NaClO) was ineffective in inactivating the ASFV in the presence of blood and similar to what was observed in the presence of feces [34]. Three studies tested different commercial products containing sodium hypochlorite against the ASFV, of which two did not report the commercial name. The first study available from Shirai et al. [41] tested the Purelox product containing 6% sodium hypochlorite (NaClO) against the Lisbon 60 strain (Table 2). The product was tested at 1:100, 1:200, 1:400; 1:800, 1:1600 and 1:3200 dilutions (*v*/*v*) in water for 30 min at room temperature [41]. The virucidal effect was observed from a 1:200 to 1:800 dilution, corresponding to the effective concentration of 0.03% to 0.0075% [41]. In the study by Juszkiewicz et al. [45], the commercial product tested based on sodium hypochlorite (NaClO) proved to be efficacious at 0.5 and 1% concentrations in low-level soiling conditions but only at 1% in high-level soiling conditions, demonstrating the negative impact of the presence of organic material on the disinfectant efficacy. Another commercial disinfectant, indicated as disinfectant B based on stabilized sodium hypochlorite (NaClO) (600 ppm), was tested against the ASFV strain BA71V by Krug and collaborators [34]. The commercial product was able to induce a decrease in the ASFV titers of four log_10_ on plastic and steel surfaces and on sealed concrete by 5 min of contact time [34]. The same product was not able to produce a four log_10_ reduction of the ASFV-positive swine blood or meat juice dried on steel following 10 min of contact time. A recent study by Rhee et al. [42] demonstrated the virucidal efficacy of electrolyzed water against the ASFV. Electrolyzed water or electrochemically activated (ECA) water is a stable chlorinated water obtained through the electrolysis of a diluted salt solution [58,59,60]. The ASFV BA71V was used for the virucidal efficacy test. Low- and high-level organic soiling conditions were simulated using hard water with and without 5% FBS. The virus−disinfectant mixture was prepared by mixing the virus in a ratio of 1:1 (*v*/*v*) with AEW containing 5, 10, 20, 40, 60, 80, 100, 120 and 140 ppm of free chlorine. This mixture was incubated for 30 min at +4 °C and then titrated in cell cultures to assess the decrease in virus titers. At low-level organic soiling conditions, 40 and 60 ppm of free chlorine were efficacious in yielding a greater than four log_10_ reduction of ASFV. At high-level organic soiling conditions, 80 and 140 ppm of free chlorine resulted in a decrease of more than four log_10_ [42].

Altogether, these data indicated that concentrations from 1% to 6% of sodium hypochlorite (NaClO), according to the test temperature, might be considered to reduce the ASFV titers of four log_10_ in experimental conditions, taking into account that the presence of organic material may decrease its virucidal activity.

#### 2.2.5. Iodine Compounds

The mode of action of iodine compounds is based on the destruction of membranes and the interaction with proteins, peptides, lipids, enzymes and sulfhydryl compounds. Iodine compounds are largely used as antibacterials or antivirals and they are safe for persons, animals and the environment.

Two studies are available on iodine compounds against the ASFV [41,51]. The first study available was by Shirai et al. [41] who tested a commercially available product (Poliup-3) containing a 3% concentration of potassium tetraglycine triiodide. The virucidal activity of the product was assessed against the Lisbon 60 strain at the following dilutions: 1:100, 1:200, 1:400, 1:800, 1:1600 and 1:3200 for 30 min at room temperature. The virucidal efficacy of the disinfectant was observed at the final dilutions of 1:200 and 1:400, corresponding to 0.0015 and 0.0075%, which were determined to be effective concentrations of the commercial product [41].

More recently, Pan and collaborators [51] tested iodine-based compounds against the ASFV strain Pig/HLJ/18 (Genbank accession number MK333180.1). In this study, a highly complexed povidone (HPCI) product was tested in comparison to a commercially available povidone−iodine (PVP-I) product. The two products differed by the level of free iodine content: HPCI relies on very low free iodine content compared to PVP-I [51]. The virucidal efficacy was tested using a suspension method and a spray method. The ASFV was incubated in a ratio of 1:10 (*v*/*v*), with different concentrations of both products (0.25, 1 and 5%) diluted in sterile hard water for 5, 10 and 30 min and then the mixture was titrated on cell cultures. The spray method used membrane pieces spotted with 15 microliters of ASFV and then sprayed twice with the different concentrations of disinfectants. Following 5, 10 and 30 min, the membrane pieces were immersed in a cell culture medium in order to recover the viable virus. Using both methods, the ASFV was completely inactivated following 5 min of contact time with 5% HPCI and following 15 min of contact time with 5% PVP-I. The lowest concentration of HPCI efficacious in completely inactivating the ASFV at the initial titer of 10^5^ TCID_50_/mL was 0.25% for 5 min by using the spray method but not the suspension method. In the same conditions, PVP-I did not completely inactivate the ASFV. By contrast, 1% PVP-I for 5 min of contact time was efficacious against the ASFV at 10^5^ TCID_50_/mL but not at higher virus titers using the spray method. In conclusion, HPCI at 0.25% concentration for 5 min of contact time completely inactivated the ASFV at 10^5^ TCID_50_/mL; the same was observed with 1% PVP-I using the spray method [51]. The studies available on iodine compounds against the ASFV showed that some commercially available products are efficacious but still, limited evidence exists that can provide suggestions and recommendations for their use against the ASFV.

#### 2.2.6. Oxidizing Agents

Oxidizing agents act as disinfectants trough the development of a free hydroxyl radical that oxidizes lipids and nucleic acids. Their efficacy is highly affected by the presence of organic material. Therefore, it is important that surfaces are cleaned prior to disinfection with these agents. They are corrosive for metals and irritating for mucus membranes, eyes and skin.

The ASFV Lisbon 60 strain was used to test the virucidal activity of the combination of heat, alkalinity, peroxide and time (HAPT) [44]. Kalmar et al. [44] demonstrated that peroxide strongly increased the sensitivity of the ASFV Lisbon 60 strain to alkalinity (pH = 10.2) and heat treatment (+48 °C) in the presence or absence of porcine plasma. Treatment of the ASFV with high (102.9 mM) and low (20.6 mM) HAPT for 10 and 20 min, respectively, were required to inactivate an initial titer of 7.12 [44]. At higher temperatures, a lower contact time was necessary for the virus inactivation maintained in MEM. The same experiment using swine plasma showed a protective effect on the virus. The sensitivity of the ASFV to alkalinity (NaOH) increased proportionally with temperatures. The ASFV was inactivated following 30 min at +48 °C by applying the following conditions: pH 10.2; peroxide 92.7 mM [44]. The efficacy of vapor-phase hydrogen peroxide (H_2_O_2_) (VPHP) using a commercial decontamination system was tested by Heckert et al. [47] on the ASFV Lisbon 61. The VPHP was vaporized starting from a 30% (*wt*/*wt*) solution of aqueous H_2_O_2_ at a rate of 2 g/min for 30 min in order to obtain and maintain a final concentration of 0.12% VPHP (1211 ppm). The authors tested the virucidal activity of VPHP on the ASFV in a liquid suspension with 5% FBS and ASFV dried on non-porous surfaces at room temperatures for 16 hours. The liquid and dried ASFV forms were exposed at VPHP in glass vials and on stainless steel coupons. Such treatment proved to reduce the ASFV titer to <10 TCID_50_/mL, demonstrating the applicability of VPHP in laboratory settings [47]. Zhang et al. [46] tested the inactivation capacity of ozonized water against the wild-type ASFV SY18 strain (WT-ASFV) and reporter ASFV (ASFV-ΔMGF-EGFP, with deletion of the MGF gene and introduction of the EGFP reporter based on the SY18 strain). The following concentrations of ozonized water were tested: 5, 10 and 20 mg/L for 1, 3, 6 and 10 min at room temperature. A reduction in ASFV titers was observed, ranging from two to three log_10_ but not four log_10_. Increasing the ozonized water concentration to 20 mg/L, the log_10_ reduction was higher (3).

Regarding the efficacy of the commercially available product Virkon against the ASFV, one study is available [39]. Several other studies are available on the same active compound contained in the commercial product Virkon [34,43,45]. Gabbert et al. [39] tested the efficacy of Virkon on porous concrete surfaces according to the OECD guidelines (Figure 3) [31]. Three different concrete formulations were tested: two industrial and one ready-made product, plus the same three formulation coupons receiving carbonation treatment to simulate the pH decrease of concrete, which was observed when exposed for a prolonged period to atmospheric conditions. In addition, stainless steel was tested as a non-porous surface. Concrete coupons were treated with the ASFV BA71V strain suspended in 5% BSA, 5% YE solution and 0.4% BM. Virkon S was tested at dilutions of 1, 2 and 5% in deionized water. Following 10 min of exposure using 1 or 2% Virkon TM S, the ASFV was inactivated completely on the stainless steel. The treatment of concrete carbonate coupons with 2 and 5% concentrations for 5 or 10 min resulted in a ASFV titer reduction to <1 log_10_. Krug et al. [34] tested a commercial product based on potassium peroxymonosulfate (K_2_S_2_O_8_) on surfaces used in pork packing plants. The ASFV was dried on porous (concrete) and non-porous (steel) surfaces (Figure 3) and then exposed to the disinfectant for 10 min. Following this contact time, virus titration of the virus−disinfectant mixture was performed. The commercial product tested was efficacious in yielding a four log_10_ reduction on all tested surfaces. Sovijit et al. [43] tested two commercial products, based on potassium hydrogen peroxymonosulfate (PHP) (H_2_SO_5_) against the ASFV strain VNUA-ASFV-L01/HN/04/19. All products were tested at the dilutions of 1:200, 1:400 and 1:800 for three contact times: 1, 5 and 30 min at two temperatures: +4 °C and +20 °C. One product based on PHP at a 1:200 dilution was effective in yielding a four log_10_ reduction of the ASFV following 30 min of exposure at +20 °C and the second product at both temperatures following 30 min of contact at a 1:200 dilution. Juszkiewicz et al. [45] tested potassium peroxymonosulfate (K_2_S_2_O_8_) at 0.5, 1 and 2% dilutions following the UNI EN 14675:2015 standard at both soiling conditions: low and high. The chemical was effective in causing a four log_10_ decrease in virus titer at dilutions of 0.5 and 1% at both soiling conditions. The 2% dilution caused toxicity to cell cultures and therefore, it was not possible to perform the virucidal test according to the suspension method [45]. The data available show that potassium hydrogen peroxymonosulfate (H_2_SO_5_) has a virucidal activity against the ASFV, being the chemical compound tested with a wider range of ASFV strains compared to other groups of chemicals (Table 2).

#### 2.2.7. Phenol Compounds

The mechanism of virucidal action of phenol compounds is specifically due to the denaturation and precipitation of proteins at a high concentration. Phenol compounds maintain their activity in the presence of organic matter and they are of low to moderate cost. They are generally safe for users, although in some cases skin irritation can occur. Two studies are available on phenol compounds against the ASFV [45,49]. The first study carried out by Stone and collaborators [49] assessed the virucidal activity of three commercially available disinfectants (Environ, Environ D and One Stroke Environ) against the ASFV Lisbon 60. The disinfectants were mixed at two different concentrations: 0.5 and 1% with a positive ASFV spleen homogenate (1:10 *v*/*v*) for 60 min at room temperature. Following the contact time, 1 mL of the virus−disinfectant mixture was inoculated intramuscularly in two swine per concentration of each disinfectant. Only one (One Stroke Environ) product proved to be safe without causing clinical signs and death in inoculated swine. In addition, the same experiment was performed to assess the minimum concentration and contact time efficacious against the ASFV. A complete virucidal activity of the One Stroke Environ was demonstrated in vivo at 0.5% and for a minimum of 60 min of contact time [49].

A more recent study tested the phenol according to the suspension method of the EU standard. The commercial product, tested at 0.5, 1 and 2%, was able to cause a four log_10_ reduction in low- and high-level soiling conditions at 1% following a contact time of 30 min [45], while the highest dilution tested induced a cytotoxic effect on cell cultures.

#### 2.2.8. Quaternary Ammonium Compounds

Quaternary ammonium compounds, commonly known as quats or QACs, are cationic surfactants (surface active agents) that combine bactericidal and virucidal (generally only enveloped viruses) activity with good detergency and, therefore, cleaning ability. They are safe for personal use and inexpensive. Shirai et al. [41] tested QACs against the ASFV Lisbon 60 strain. In detail, a commercial product (Cleakil-100) was tested at different concentrations from 1:400 to 1:12,800 and mixed in suspension with the ASFV. Following 1, 5, 10, 30 and 60 min at room temperature, the disinfectant−virus mixture was titrated on cell cultures. The QAC compound was effective at a dilution of 1:3200 with effective concentrations of 0.003%. More recently, Sovijit et al. [43] tested one commercial product based on QAC compounds at three dilutions 1:200, 1:400 and 1:800 for 3 contact times: 1, 5 and 30 min at two temperatures: +4 °C and +20 °C. The QAC-based product was effective at +4 °C following 1 min and at +20 °C following 30 min of contact time at a dilution of 1:200. A recent study by Taesuji and collaborators [50] has similarities to the work of Sovijit et al. [43] regarding the ASFV strain used and the choice of didecyldimethylammonium chloride that belongs to QACs. Unlike the work of Sovijit et al., in this suspension test, five commercially available disinfectants were tested. The disinfectants, produced in Thailand, contained multiple active ingredients in order to enhance the synergistic virucidal activity. Two out of five disinfectants contained glutaraldehyde and one acetic acid in addition to didecyldimethylammonium chloride. The study showed that a 1:200 (*v*/*v*) dilution of the disinfectants tested at +4 °C and +20 °C for 30 min led to a reduction of about four log_10_. A commercially available disinfectant was recently tested on surfaces used for pork packing plants [34]. In detail, the test method applied was a modification of the ASTM E1053-20 method [30] (Figure 2). The ASFV (BA71V) was dried on steel, plastic and sealed concrete surfaces and subsequently subjected to disinfection for 10 min. The commercial product based on QACs was able to cause a 3.8 log_10_ titer reduction on steel, a 3.9 log_10_ titer reduction on plastic and a 4.1 log_10_ titer reduction on concrete [34]. The reduction of the ASFV titer on stainless steel surfaces was reached after 5 min of exposure to the disinfectant [34]. A different commercial product based on benzalkonium chloride was tested recently [45] according to the UNI EN 14675:2015 standard (Figure 1). The product was tested against the ASFV BA71V at three dilutions: 0.5, 1 and 2% and in both soiling conditions (low and high). Only a 1% dilution caused a decrease of 4.25 log_10_ in virus titers in the low-level soiling condition [45]. At the highest dilution, the cytotoxic effect did not enable the assessment of the virucidal activity of the product.

#### 2.2.9. Plant Extracts

Fourteen different plant extracts were recently tested according to the UN ENI standard at low- and high-level soiling conditions at three concentrations: 30, 60 and 80% for 30 min against the BA71V ASFV strain at +10 °C for 30 min. Only peppermint extract (*Mentha piperita*) produced a four log_10_ reduction at a 30% active concentration [52].

## 3. Discussion

The last and ongoing ASF epidemic wave has highlighted the difficulties in controlling the infection in both developed and in-transition countries. Direct and indirect transmission through contact with infected domestic and wild animals and fomites are the major risks of the introduction and secondary spread of the disease. In the absence of vaccination tools, the prevention of the introduction and the secondary spread of the ASFV in the domestic pig industry strictly relies on biosecurity measures.

Biosecurity is defined in several ways by international bodies such as the OIE [61] and FAO [62] and presents a precise definition according to the Animal Health Law [63]. All the definitions available describe biosecurity as a set of measures, procedures and attitudes that have the common goal of reducing the risk of the introduction and spread of the disease. An assessment of risk factors for the introduction and secondary spread of the ASFV is therefore unavoidable to identify effective biosecurity measures that rely on: segregation, cleaning and disinfection. The decontamination therefore becomes one of the major mitigation procedures necessary to implement and properly plan. Decontamination is a complex process that is primarily dependent on the availability of baseline information on the physical and chemical resistance of the virus and secondly, on characteristics of the premise/environment to be decontaminated, climate conditions at the time of decontamination, cost of the decontamination, availability of good quantities of efficacious products and its safety for personnel. In this view, the standardization of the decontamination process remains challenging.

The present review summarized reports available on the efficacy of chemical compounds and commercial disinfectants against the ASFV, highlighting some aspects that deserve attention for future studies. Most of the studies available adopted international standards and guidelines in order to investigate the virucidal activity of substances under study [19,22,23,24,25,26,27,28,29], rendering data comparable and reproducible. However, of note is that the majority of the studies available on the virucidal efficacy of products against the ASFV adopted the suspension method according to the UNI EN 14675:2015. This method is far from reproducing a field condition where the virus is not in suspension but, rather, associated to body fluids, tissues and excretions. In fact, according to the UNI EN 14675:2015 standard, the presence of the organic material is approximated by the use of BSA and YE as interfering substances that may decrease the disinfectant activity. By contrast, standards based on the carrier test method better mimic a field condition, as the virus under study is associated to a carrier.

Regarding available data on the virucidal activity of chemical compounds against the ASFV adopting the carrier test methods on non-porous surfaces (i.e., ASTM and OECD), only four studies are available [33,39]. In particular, only one study is available on porous surfaces supporting the necessity to increase the number of studies on different types of surfaces [39]. These studies tested three types of surfaces: plastic, steel and concrete and chemicals belonging to chlorine compounds, oxidizing agents, QACs and acids, suggesting that further studies might be generated to test additional compounds and types of surfaces and to validate data generated using other ASFV strains [32,33,39].

Regarding the variety of ASFV strains used to test the virucidal activity of chemical compounds, it emerged that these are limited to the use of genotype I and Vero-adapted ASFV strains such as the BA71V and Lisbon 60 strains. Few studies tested field ASFV strains [24,26,27,30,33,35] belonging to genotype I and II and only one study used a field ASFV belonging to genotype VIII [40]. Regarding the virulence and hemadsorption characteristics of the ASFV tested, no data is available. This highlights that no information is available using field ASFV strains and currently circulating genotype II in Europe, genotype I and endemic genotypes in Africa. An explanation for the restricted variability in testing various ASFV strains is the non-easy cultivability of the virus, which impacts the selection of ASFV strains and the production of high titers of virus stocks necessary to prove the decrease in virus titers and therefore, the disinfectant efficacy. This represents one of the major bottlenecks in applying available international guidelines for testing the virucidal activity of chemicals against the ASFV.

The most tested products against the ASFV were chlorine compounds, oxidizing agents that are recommended by the OIE manual on the ASFV [61]. Both groups of chemicals are highly dependent on the absence of organic material in order to be highly efficacious. Therefore, a thorough cleaning is needed when these compounds are selected.

Few studies were carried out on aldehydes, phenol compounds and iodine compounds. In particular, phenol compounds appear to be efficacious in the presence of organic material and are low cost [54]. In addition, no data are available on the virucidal activity of alcohols against the ASFV. Data available regarding the efficacy of chemical compounds against the ASFV indicate that all groups of chemicals tested are efficacious against the ASFV, differing on the mode of action and the applicability on different materials. Based on the data available, sodium hypochlorite has an excellent efficacy against the ASFV, but its efficacy decreases in the presence of organic material; it can be irritating and corrosive, limiting its application in several scenarios. Oxidizing agents are powerful compounds. During decontamination procedures, they present corrosiveness for several metals and this should be taken into account when planning the decontamination of infected pig holdings. Glutaraldehyde is widely used in the veterinary sector, but it is irritating for the skin and eyes. However, few studies are available on its efficacy against the ASFV and further investigations in different experimental settings may clarify its efficacy and proposed use during ASFV outbreaks.

Phenol compounds may have a good range of applications. They seem to be effective against the ASFV and are still efficacious in presence of organic material. Regarding the decontamination of clothing, acids and QACs may be used as they are safe and efficacious against the ASFV. The present review highlights that gaps in the knowledge still exist on the efficacy of chemical compounds against the ASFV. Further studies are needed to validate data on the efficacy of tested chemical compounds in different experimental settings to investigate the efficacy of additional compounds and their application. The methodologies applied so far to investigate the virucidal activity of chemicals are limited to laboratory experiments that do not properly reflect the field situation. Additional laboratory protocols should be developed, aiming to simulate the variety of ASFV epidemiological scenarios. ASFV is a complex virus causing a complex disease with two epidemiological cycles (i.e., domestic and sylvatic) that require different approaches from a disease management perspective. This is reflected even in the decontamination phase of the ASF that, therefore, may be planned and developed in different ways according to the field scenario. This implies that ad hoc decontamination protocols and procedures may be needed. The availability of data on the virucidal activity of chemicals and their use against the ASFV generated under laboratory conditions developed to simulate the variety of field contexts in which the ASF may emerge are crucial tools to manage ASF outbreaks. This would support all the practicalities of disinfecting infected swine premises with the ASFV.

National and international organizations involved in controlling the ASFV should promote, in a coordinated manner, the collection and development of guidelines and protocols to support the effected countries in managing the critical aspects of decontamination.

## Figures and Tables

**Figure 1 viruses-14-01384-f001:**
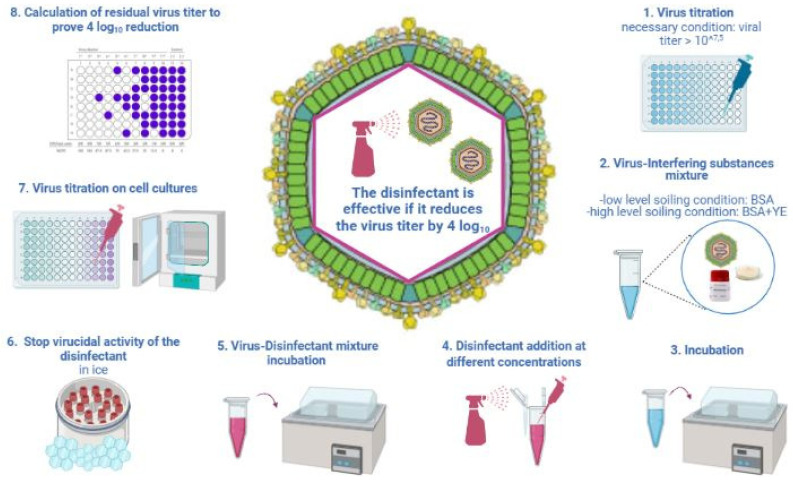
Quantitative suspension test for virucidal activity against ASFV according to the UNI EN 14675:2015 standard. BSA: bovine serum albumin; YE: yeast extract. Created in Biorender.com (https://biorender.com/), last accessed on 8 March 2022.

**Figure 2 viruses-14-01384-f002:**
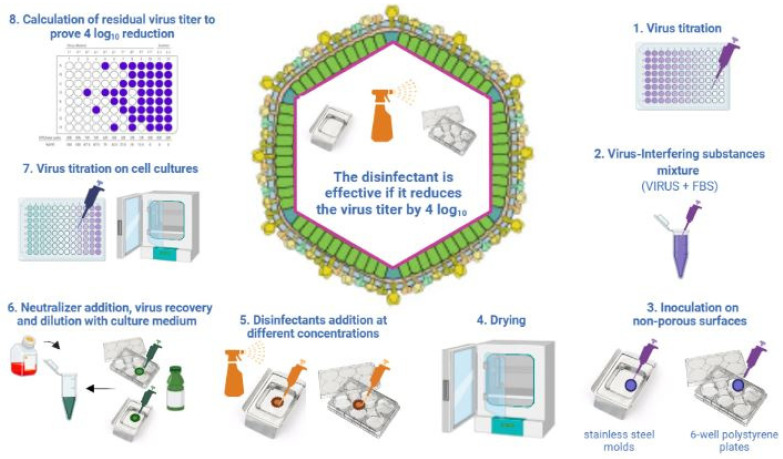
Test method applied to assess the virucidal activity of disinfectant on inanimate non-porous environmental surfaces according to ASTM E1053-20 standards. The figure reports the method adopted by Krug et al. [32] for ASFV. Created in Biorender.com (https://biorender.com/), last accessed on 8 March 2022. FBS: fetal bovine serum.

**Figure 3 viruses-14-01384-f003:**
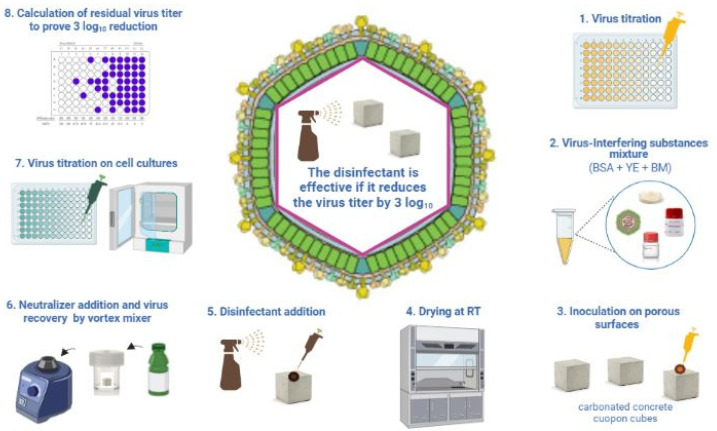
Test method applied to assess the virucidal activity of disinfectant on hard non-porous surfaces according to OECD guidelines adopted with slight modification by Gabbert et al. [39] for porous surfaces. Created in Biorender.com (https://biorender.com/), last accessed on 8 March 2022. BSA: bovine serum albumin; YE: yeast extract; BM: bovine mucin.

**Table 1 viruses-14-01384-t001:** Comparison of conditions of the international standards and guidelines adopted to test the virucidal activity of chemical compounds against ASFV.

	UNI EN 14675:2015 Quantitative Suspension Test (Phase 2/Step 1)	OECD 2013 Carrier Test (Disks of Stainless Steel)	ASTM E1053-20 Carrier Test (Glass Petri Dishes)
**Virus Volume**	1000 µL	10 µL	200 µL
**Disinfectant Volume**	8000 µL	50 µL	2000 µL
**Ratio**	1:8	1:5	1:10
**Interfering Substances**	1% BSA + 1% YE OR ONLY 0.3% BSA	5% YE + 5% BSA + 5% BM	NOT REQUIRED
**Contact Time and Temperature**	30 min ± 10 s + 10 °C ± 1 °C	LABEL INDICATION + 20–25 °C	LABEL INDICATION + 20–25 °C
**Water**	400 ppm	338–394 ppm	400 ppm
**Virus Titer Reduction**	4 log_10_	3 log_10_	4 log_10_

ppm: parts per million; min: minutes; s: seconds, BSA: bovine serum albumin; YE: yeast extract; BM: bovine mucin.

**Table 2 viruses-14-01384-t002:** List of tested and efficacious disinfectants against ASFV.

Chemical Group/Active Substance	Active Concentration	Contact Time (min)	Temperature (°C)	Virus	Genotype	Conditions	Test Method	Paper	Recommended Use
**Alkalis**									
Sodium hydroxide	1%	5	4	Lilongwe 20/1	VIII	Pig slurry	Suspension	Turner et al., 1999 [40]	Not efficacious at room temperature. Do not use in the presence of aluminium and derived alloys
	2 and 3%	30	10	BA71V	I	BSA; BSA + YE	UNI EN 14675:2015	Juszkiewicz et al., 2020 [45]	
Calcium hydroxide	1%	5	4	Lilongwe 20/1	VIII	Pig slurry	Suspension	Turner et al., 1999 [40]	Use for walls and floors
**Acids**									
Acetic acid	1	10	22	BA71V	I	Steel and plastic	ASTM E1053-20 modified	Krug et al., 2011 [32]	
	2%	30	10	BA71V	I	BSA; BSA + YE	UNI EN 14675:2015	Juszkiewicz et al., 2020 [45]	
Citric acid	1 and 2%	10	22	BA71V	I	Plastic	ASTM E1053-20 modified	Krug et al., 2011 [32]	Safe for clothes and body decontamination
	1%	10	22	BA71V	I	Steel	ASTM E1053-20 modified	Krug et al., 2011 [32]	
	2%	30	22	BA71V	I	Birch wood veneer	ASTM E1053-20 modified	Krug et al., 2012 [33]	
**Chlorine Compounds**									
Sodium hypoclorite	500 ppm	10	22	BA71V	I	Steel and plastic	ASTM E1053-20 modified	Krug et al., 2011 [33]	Effective for most applications, decreased efficacy in presence of organic material. Less stable in warm, sunny conditions above +15 °C. Toxic for eyes and skin
	2000 ppm	30	22	BA71V	I	Birch wood veneer	ASTM E1053-20 modified	Krug et al., 2012 [33]	
	6%	30	RT	Lisbon 60	I	None	Suspension	Shirai et al., 2000 [41]	
	1%	30	10	BA71V	I	BSA, BSA + YE	UNI EN 14675:2015	Juszkiewicz et al., 2020 [45]	
Acidic electrolyzed water	80 ppm	30	4	BA71V	I	5% FBS	Suspension	Rhee et al., 2021 [42]	
**Oxiding Agents**									
Ozonized water (O3)	20 mg/L	10	20–25	SY 18	II	None	Suspension	Zhang et al., 2020 [46]	
Potassium hydrogen	600 ppm	10	RT	BA71V	I	Steel, plastic, concrete	ASTM E1053-20 modified	Krug et al., 2018 [34]	Use for laboratory equipment. Excellent disinfectant active against all viruses and bacteria. Mildly corrosive for many metals
	1/200	30	20	VNUA-ASFV-L01/HN/04/19	II	None	Suspension	Sovijit et al., 2021 [43]	
	1/200	30	4 and 20	VNUA-ASFV-L01/HN/04/19	II	None	Suspension	Sovijit et al., 2021 [43]	
	1%	30	10	BA71V	I	BSA, BSA + YE	UNI EN 14675:2015	Juszkiewicz et al., 2020 [45]	
	2 and 5%	5, 10	20–25	BA71V	I	BSA + YE + BM	OECD 2013	Gabbert et al., 2020 [39]	
Vaporized hydrogen peroxide	30%	30	30–40	Lisbon 61	I	5% FBS	Vaporization	Heckert et al., 1997 [47]	Use for laboratory equipment
Hydrogen peroxide	102.6 mM (35% stock solution)	10	48	Lisbon 60	I	Plasma	Suspension	Kalmar et al., 2018 [44]	Use for laboratory equipment. Rinse after use
**Aldehydes**									
Glutaraldehyde	0.1%	30	10	BA71V	I	BSA + YE	UNI EN 14675:2015	Juszkiewicz et al., 2019 [48]	Excellent disinfectant effective against all viruses and bacteria. Avoid eye and skin contact
	1%	30	10	BA71V	I	BSA	UNI EN 14675:2015	Juszkiewicz et al., 2020 [45]	
**Phenol Compounds**									
Phenol	1%	30	10	BA71V	I	BSA + YE	UNI EN 14675:2015	Juszkiewicz et al., 2020 [45]	Efficacious in the presence of organic material. Rinse after use
o-Phenilphenol	0.5%	60	RT	Lisbon 60	I	None	In vivo test	Stone and Hess 1973 [49]	
**Quaternary Ammonium Compounds**									
Benzalkonium chloride	1%	30	10	BA71V	I	BSA	UNI EN 14675:2015	Juszkiewicz et al., 2020 [45]	Recommended for pesonal use. Do not use with hard water
Quaternary ammonium	1/200	1	4	VNUA-ASFV-L01/HN/04/19	II	None	Suspension	Sovijit et al., 2021 [43]	
Didecyldimethylammonium chloride	10%	30	RT	Lisbon 60	I	None	Suspension	Shirai et al., 2000 [41]	
	0.09%-0.0275%/0.1%	30	4 and 20	VNUA-ASFV-L01/HN/04/19	II	None	Suspension	Taesuji et al., 2021 [50]	
Quaternary ammonia	800 ppm	10	RT	BA71V	I	Steel, plastic, concrete	ASTM E1053-20 modified	Krug et al., 2018 [34]	
**Iodine Compounds**									
Povidone-iodine (5% iodine content)	5%	15	RT	ASFV pig/HLJ/18	II	None	Spray	Pan et al., 2021 [51]	
Potassium tetraglicine triiodide	3%	30	RT	Lisbon 60	I	None	Suspension	Shirai et al., 2000 [41]	
**Plant Extracts**									
Peppermint	30%	30	10	BA71V	I	BSA; BSA + YE	UNI EN 14675:2015	Juszkiewicz et al., 2021 [52]	

Suspension: the method adopted that resembled the UNI EN 14675:2015, putting the cell-cultured ASFV in a liquid form in contact with the tested compound.

## Data Availability

No new data were created or analyzed in this study.

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
