# Peer review of "Disinfectants against African Swine Fever: An Updated Review"

_viruses, 2022, doi:10.3390/v14071384_

Round 1
Reviewer 1 Report
In this manuscript, the authors comprehensively summarized the disinfectants against ASF. The review is well organized, and I only have a few minor suggestions to improve the manuscript.
- I would suggest the authors explain the abbreviations used in Table 1 in the footnote.
- I would suggest the authors add some information about the ASFV virology, especially the different genotypes, as it is directly related to the information in the main text and Table 2.
Author Response
Dear Editor,
We thank the reviewers for comments and suggestions provided in order to render the paper more readable and useful for the scientific community. We have carefully addressed all the comments of both reviewers modifying the review. We hope this version now meets the quality standards of Viruses.
Responses to reviewers are provided after each comment in italics. Revisions have been highlighted using the track-changes tool in the revised version of the manuscript.
REVIEWER 1
Comments and Suggestions for Authors
In this manuscript, the authors comprehensively summarized the disinfectants against ASF. The review is well organized, and I only have a few minor suggestions to improve the manuscript.
I would suggest the authors explain the abbreviations used in Table 1 in the footnote.
R: abbreviations have been explained in the footnote of Table 1.
I would suggest the authors add some information about the ASFV virology, especially the different genotypes, as it is directly related to the information in the main text and Table 2.
R: This information has been added in the introduction section.
Reviewer 2 Report
This review is timely as ASFV has been on the move and without a vaccine or therapeutic, biosurveillance and strategies to decontaminate infected premises are currently the only “shovel ready” responses available. Decontamination of production units, transportation vehicles and processing plants will be a major task in the event of widespread outbreak of ASFV. As the authors have stated there are limited disinfectant efficacy experiments that mimic real live situations reported in the literature. In addition, they also highlight the lack of an international validated standard protocol for testing disinfectants, thus highlighting the need for further experimental activity in this space. The authors provide a comprehensive overview of the modes of action of each of the different families of disinfectants that have been tested for their potency for the inactivation of ASFV. The authors do not address the impact of the various chemicals if released into the environment, specifically if they are leached into groundwater. This aspect has to be considered to keep within limits set by national environmental agencies.
There are many typographical errors and there are sentences and phrases that have failed in translation. Below is a list of what caught my eye. The discussion paragraphs are too long and don’t give the reader a break to consider the points.
Line 31 …precisely.. does not make sense
37 invading the European Union …. “into European Union”
43 AFSV… ASFV
59 are useful… provides useful (reads better)
79 “antiviral” disinfectants should be changed to “virucidal”
114 disinfect… disinfectant
115 re-do this sentence -grammar, spelling and structure
The use of “caused” a X log reduction – suggest use resulted in or yielded a 4 log reduction. Caused has been used in a number of occasions, please substitute
266 and 439 tire …. titre or titer depending on style
294 aortic valvae should this not be valve
321 hypocloric acid … HOCl or hydrochloric acid (HCL)????
330 ..determined a 4 log… demonstrated, resulted, yielded
343 ..similarly ………. “ similar to what was observed”
364 mimic … simulating
449 testes .. tested
628 restructure this sentence
637 request .. require?
644 and .. an
-Table 1, line 95, insert a pagebreak before the header, redo the table, instead of Variable maybe “Constituents” would be a better fit remove Test Type from under the left header.
-Table 2: they have electrolyzed water under the chlorine compound section?
This is a good review of the current literature in the ASF decontamination/disinfectant research space, it needs a little work with grammar and spellings.
Author Response
Dear Editor,
We thank the reviewers for comments and suggestions provided in order to render the paper more readable and useful for the scientific community. We have carefully addressed all the comments of both reviewers modifying the review. We hope this version now meets the quality standards of Viruses.
Responses to reviewers are provided after each comment in italics. Revisions have been highlighted using the track-changes tool in the revised version of the manuscript.
REVIEWER 2
Comments and Suggestions for Authors
This review is timely as ASFV has been on the move and without a vaccine or therapeutic, biosurveillance and strategies to decontaminate infected premises are currently the only “shovel ready” responses available. Decontamination of production units, transportation vehicles and processing plants will be a major task in the event of widespread outbreak of ASFV. As the authors have stated there are limited disinfectant efficacy experiments that mimic real live situations reported in the literature. In addition, they also highlight the lack of an international validated standard protocol for testing disinfectants, thus highlighting the need for further experimental activity in this space. The authors provide a comprehensive overview of the modes of action of each of the different families of disinfectants that have been tested for their potency for the inactivation of ASFV. The authors do not address the impact of the various chemicals if released into the environment, specifically if they are leached into groundwater. This aspect has to be considered to keep within limits set by national environmental agencies.
There are many typographical errors and there are sentences and phrases that have failed in translation. Below is a list of what caught my eye. The discussion paragraphs are too long and don’t give the reader a break to consider the points.
R:The paper has been revised and modified to avoid errors and many sentences rephrased.
Line 31 …precisely.. does not make sense
R: deleted
37 invading the European Union …. “into European Union”
R: According to the Oxford dictionary is correct as it is written.
43 AFSV… ASFV
R: modified and checked throughout the document.
59 are useful… provides useful (reads better)
R: modified accordingly.
79 “antiviral” disinfectants should be changed to “virucidal”
R: modified accordingly.
114 disinfect… disinfectant
R: modified accordingly.
115 re-do this sentence -grammar, spelling and structure
R: the following sentence: “This method is adopted to registry chemicals as virucidal in the EU and the virus that must be tested for such purpose is the Bovine Enterovirus 1 [16]. The use of other viruses is not compulsory for registration purposes of virucidal disinfectants.” has been modified in the following one:
“Therefore products that must be registered in EU as virucidal disinfectants must be tested following this method and against Bovine Enterovirus 1. The use of other viruses is not optional and not required for registration purposes.”
The use of “caused” a X log reduction – suggest use resulted in or yielded a 4 log reduction. Caused has been used in a number of occasions, please substitute
R: modified accordingly.
266 and 439 tire …. titre or titer depending on style
R: all the document has been checked to substitute tire in titer and titre in titer according to British English.
294 aortic valvae should this not be valve
R: modified accordingly.
321 hypocloric acid … HOCl or hydrochloric acid (HCL)????
R: modified in HOCl.
330 ..determined a 4 log… demonstrated, resulted, yielded
R: modified in resulted.
343 ..similarly ………. “ similar to what was observed”
R: modified as suggested.
364 mimic … simulating
R: modified accordingly.
449 testes .. tested
R: modified accordingly.
628 restructure this sentence
R: The following sentence: “Notwithstanding the available of a good piece of information on efficacy of chemicals against ASFV, the present review highlights that gaps of knowledge still exist on the re-sistance of ASFV to chemical compounds”.
Has been modified in the following one:
“The present review highlights that gaps of knowledge still exist on the efficacy of chemical compounds against ASFV to chemical compounds.”
637 request .. require?
R: modified accordingly.
644 and .. an
R: modified accordingly.
-Table 1, line 95, insert a pagebreak before the header, redo the table, instead of Variable maybe “Constituents” would be a better fit remove Test Type from under the left header.
R: modified according to the reviewer suggestions.
-Table 2: they have electrolyzed water under the chlorine compound section?
R: Hypochloric acid (HOCl) is a form of free chlorine and an active component of electrolyzed water, and therefore can be classified under both chlorine compound or oxidising agent.
This is a good review of the current literature in the ASF decontamination/disinfectant research space, it needs a little work with grammar and spellings.
R: the paper has been revised for grammar and spellings